# Effects of Vinyl Functionalized Silica Particles on Thermal and Mechanical Properties of Liquid Silicone Rubber Nanocomposites

**DOI:** 10.3390/polym15051224

**Published:** 2023-02-28

**Authors:** Yulong Zhang, Wei Liu, Qiang Zhou, Yiting Meng, Ye Zhong, Jing Xu, Chuan Xiao, Guangpu Zhang, Yanan Zhang

**Affiliations:** 1China North Industry Advanced Technology Generalization Institute, Beijing 100089, China; 2Ordnance Science and Research Academy of China, Beijing 100089, China; 3National Special Superfine Powder Engineering Research Center of China, Nanjing University of Science and Technology, Nanjing 210014, China; 4College of Materials Science and Engineering, Nanjing Tech University, Nanjing 211816, China

**Keywords:** surface modification, silica, thermal stability, thermal conductivity, mechanical properties

## Abstract

It is very important to develop a new method of preparing high-performance liquid silicone rubber-reinforcing filler. Herein, the hydrophilic surface of silica (SiO_2_) particles was modified by a vinyl silazane coupling agent to prepare a new type of hydrophobic reinforcing filler. The structures and properties of modified SiO_2_ particles were confirmed using Fourier-transform infrared spectroscopy (FT-IR), X-ray photoelectron spectrometer (XPS), specific surface area and particle size distribution and thermogravimetric analysis (TGA), the results of which demonstrated that the aggregation of hydrophobic particles is greatly reduced. Additionally, the effects of the vinyl-modified SiO_2_ particle (*f*-SiO_2_) content on the dispersibility, rheology, and thermal and mechanical properties of liquid silicone rubber (SR) composites were studied for application toward high-performance SR matrix. The results showed that the *f*-SiO_2_/SR composites possessed low viscosity and higher thermal stability, conductivity, and mechanical strength than of SiO_2_/SR composites. We believe that this study will provide ideas for the preparation of high-performance liquid silicone rubber with low viscosity.

## 1. Introduction

Liquid silicone rubber (SR) is a type of organic–inorganic hybrid material and possess many advantages that cannot be replaced by other materials [1]. Compared to ordinary organic rubbers, it has better thermal stability, chemical stability, electrical insulation, and lower surface tension [2]. Based on these unique performances, silicone rubber is widely used in the aerospace, military, construction, electrical, medical, and other industries [3,4]. However, the research on its thermal and mechanical enhancement has never stopped [5,6,7]. The addition of inorganic fillers and chemical modification of the polysiloxane matrix are two effective methods of enhancing the properties of polymers [7,8,9].

Inorganic fillers, including silica particles (SiO_2_), are commonly found to be a simple and effective way to improve thermal stability and mechanical properties, such as tensile and compressive strengths [10,11,12,13]. However, the surface of SiO_2_ is usually rich in hydroxyl groups (Si-OH), which tend to adsorb moisture from the air and cause agglomeration, further leading to the degradation of SR composite properties [14,15]. Currently, surface modification has been validated as an effective method of solving agglomeration problems and improving the affinity between silica particles and the matrix [16,17]. At present, the surface modification methods of SiO_2_ can be divided into two categories: ex situ and in situ modification. For the in situ modification method, typically, Rangsunvigit et al. prepared the modified SiO_2_ through ionic interaction between the SiO_2_ particles with two surfactants. It was effectively demonstrated that the improved affinity between rubber and modified silica enhanced the mechanical properties of rubber [18]. An et al. prepared modified hydrophobic SiO_2_ through a hydrogen bonding interaction between hydroxyl groups on the surfaces of SiO_2_ particles and hydroxyl groups of alcohols [19]. For the ex situ modification method, typically, the most commonly used modifier is hexamethyldisilazane (HMDS), which contributes to the formation of effective chemical bonding between the Si-OH groups on the surface of SiO_2_ particles and the hydrolyzable silicon–nitrogen bond of HMDS [20,21]. All of the above modification methods can increase the dispersion of SiO_2_ particles in SR matrix, thus contributing to the improvement the performance of SR in different degrees. However, the interface force between SR and the SiO_2_ particles obtained by these methods is only the weak van der Waals force, not the strong chemical bond force. Recently, Park et al. reported a new modification method of SiO_2_ particles [22]. The modified SiO_2_ particles were prepared by introducing vinyl groups to the surface of SiO_2_ particles with vinyl-containing silane coupling agent, which can improve the performance of SR via in situ polymerization with SR molecular chains. However, the effect of modifier on BET specific surface area, particle size distribution, and the rheology of the uncured composites before and after particle modification has not been systematically described, which is very important for applications. Additionally, the results show that a large amount of nanosilica (30 wt.%) is needed to obtain high-tensile-strength (6.15 MPa) silicone rubber, but the high content of added nanofillers may imply the poor fluidity of the blend.

On the basis of previous studies, the effects of SiO_2_ particles modified by tetramethyldivinyldisilazane (TMDVS) on the BET specific surface area, morphology, particle size distribution, and SR enhancement of SiO_2_ particles were systematically studied. This study revealed the relationship between modification methods, particle characteristics, and properties of composites, which is of great significance to the study of reinforcement mechanism of fillers. We also believe that this work will provide a new perspective for the design and preparation of high-performance polymer composites.

## 2. Experimental

### 2.1. Materials

The polydimethyl-vinyl siloxane (PDVS; vinyl content = 0.14%; 10,000 mPa·s) was purchased from Zhonglan Chengguang Co., Hangzhou, China. Vinyl MQ resin (VMQ, vinyl content = 1.28%; 6500 mPa·s) was provided by Zhejiang Runhe Co., Hangzhou, China. The crosslinking agent (polymethylhydrosiloxane (PHDS); hydrogen content = 0.5%; 80 mPa·s) was purchased from Shandong Dayi Co., Yantai, China. Silicone rubber (SR) was prepared by mixing PDVS and VMQ in proportion (Si-H/CH=CH_2_ = 1.5). The Karstedt’s catalyst (3000 ppm) was purchased from Zhonglan Chengguang Co., Ltd. The tetramethyldivinyldisilazane (TMDVS, China, purity 99.8%) was purchased from Maclean. The silica (SiO_2_) was purchased from Cabot (the specific surface = 250 cm^2^/g, ⍴ = 30 g/L). The ethanol (99.9%) was purchased from Beijing Chemical Works, Beijing, China and the deionized water was manufactured in our lab.

### 2.2. Surface Modification of Silica Particles

In order to modify the hydrophilic surface of silica particles (SiO_2_) and obtain hydrophobic properties, the TMDVS was used to treat the surface. Firstly, the TMDVS was hydrolyzed under acidic conditions. Amounts of 2 g TMDVS, 50 g, ethanol and 2 g ammonia solution were added into a three beaker, and the mixture was kept stirring for 2 h at 25 °C to completely hydrolyze the TMDVS. Then, 5 g SiO_2_ was added into the hydrolysate obtained from the first step, and the mixture was dispersed by ultrasound for 2 h at 80 °C. The ultrasonic energy can efficiently remove microbubbles on the surface of the SiO_2_, which facilitated the complete reaction between hydrolysate and SiO_2_ [22]. Finally, after the coupling reaction was completed, the modified SiO_2_ was washed three times with ethanol and then freeze dried to remove the solvent, and the modified silica was labeled as *f*-SiO_2_. The modification process of SiO_2_ by TMDVS is shown in Figure 1a.

### 2.3. Preparation of SR Nanocomposites

The *f*-SiO_2_ particles were dispersed in liquid PDVS and VMQ using a kneading machine to prepare SR nanocomposites. The mixing ratios of the *f*-SiO_2_ in the SR composites were 2, 4, 6, 8, and 10 wt.%, respectively. After dispersion of the *f*-SiO_2_ for 6 h, the curing agent PHDS, reaction inhibitor and Karstedt’s catalyst were sequentially mixed. After mixing evenly, the mixture was placed in a vacuum box for defoaming. Finally, the SR/*f*-SiO_2_ compound was poured into a PTFE mold and cured at 80 °C for 2 h and further cured at 120 °C for 3 h. The typical cured mechanism of in situ polymerization between *f*-SiO_2_ and SR matrix is shown in Figure 1b. The SR nanocomposites prepared for nonfunctionalized silica were fabricated in the same way as the SR/*f*-SiO_2_ nanocomposites. 

### 2.4. Characterization

The Fourier-transform infrared spectroscopy (FT-IR) of the modified SiO_2_ particles was determined using a VERTEX 70 spectrometer (Bruker, Ettlingen, Germany) in range of 400–4000 cm^−1^. The surface elemental chemical state of SiO_2_ particles was analyzed using an X-ray photoelectron spectrometer (XPS, EscaLab 250Xi, Thermo Fisher Scientific, Waltham, MA, USA) with a monochromatic Al Kα X-ray source, and the acquired data were quantified and analyzed using the XPSPEAK 4.1 program. The surface morphology of SiO_2_ particles was observed using a cold-field emission scanning electron microscope (FE-SEM, S4800, Hitachi, Japan) with an acceleration voltage of 15 kV after being sputtered with a thin layer of gold. The BET surface areas are determined by N_2_ physisorption at liquid N_2_ temperature on a Miraesi KICT-SPA 3000 instrument. The particles’ size distribution was tested using a Mastersizer 2000 Laser Particle Sizer (Malvern, Malvern, UK) in ethanol dispersion phase. TG analysis of the particles and composites was performed with a heating range from 30 to 1000 °C at the heating rate of 10 K/min under argon atmosphere. The grafting ratio of the modified product was calculated by the following Equation (1).
(1)$$G_{f}=[\left(W_{1}-W_{0}\right)/W_{0}]\times 100\%$$

In the formula, $G_{f}$ is the grafting rate of the modified product (%), $W_{1}$ is the mass of the product after grafting (g), and $W_{0}$ is the initial mass before modification (g).

The rheological characteristic of liquid *VP*-PBSi/SR (system without crosslinking agent) was conducted on a rheometer (MCR92, Anton Paar) with a shear rate between 0.01 and 600 s^−1^. The dispersion state of the modified SiO_2_ particles in SR matrix was observed using a high-resolution transmission electron microscope (HR-TEM, Hitachi HF5000) with 100–120 nm thickness at an accelerating voltage of 200 kV. The thermal conductivities of composites were measured through transient plane sources method using a thermal analyzer (TPS 2500S, Hot Disk, Sweden) with a sensor diameter of 3 mm placed between two identical samples of Φ20 × 5 mm at room temperature. The tensile property of all samples was tested using an electronic universal drawing machine (AGS-X, Shimadzu) at tensile rate of 100 mm/min with a dumbbell-like specimen. The compressive property was tested according to GB/T 7757-2009, and the samples’ specification was Φ29 mm × 12.5 mm. The hardness was tested according to GB/T 531.1-2008. The crosslinking density of the composites was calculated using the following Equations (2) and (3) [23].
(2)$$\varphi =\left(w_{s}-w_{o}\right)/\left[\left(w_{s}-w_{o}\right)/\rho_{1}+w_{o}/\rho \right]$$
(3)$$y_{e}=\rho /M_{C}=-\left[\ln\left(1-\varphi \right)+\varphi +X_{1}\varphi^{2}\right]/\left(V_{o}\varphi^{1/3}\right)$$

In the formula, φ is the percentage of the volume of rubber before swelling to the volume after swelling (%), $w_{s}\;$is the weight after swelling (g), $w_{o}$ is the weight before swelling (g), $\rho_{1}$ is the density of CCl_4_ (1.6 g/cm^3^), *ρ* is the density of rubber before swelling (g/cm^3^), and $y_{e}$ is the crosslinking density. *M_C_* is the average molecular weight between crosslinking points. $X_{1}$ is the Flory–Huggins parameter, and the value is 0.45. $V_{o}$ is the molar volume of CCl_4_ (96.5 cm^3^/mol).

## 3. Results and Discussion

### 3.1. Surface Chemical Structure Analysis of f-SiO_2_

The FI-IR spectra are shown in Figure 2a. In the spectrum for the *f*-SiO_2_ particles, typically, a new peak caused by the Si-CH=CH_2_ bond appears at 1405 cm^−1^ [22]. The peaks appearing from 2950–2800 cm^−1^ are attributed to the stretching vibration of the C-H from the CH/CH_2_/CH_3_ group [24]. The absorption peak at 1630 cm^−1^ is the bending vibration of H-O-H, and the peaks at 3455 and 800 cm^−1^ are ascribed to the existence of Si-OH. The peaks at 1132 cm^−1^ are caused by the Si-O-Si bonds [25].

To further confirm the structure of the modified SiO_2_, XPS is used to analyze the chemical elements on the surface of the silica. The wide-scan survey spectra of SiO_2_ and *f*-SiO_2_ are presented in Figure 2b. Bonding energies of silicon and carbon are centered at around 102.6 and 281.4 eV, respectively. It should be noted that only the weak peak of carbon is detected on the surface of SiO_2_. After modifying by TMDVS, the distinct XPS peak of carbon appears, which is attributed to successful surface modification. Moreover, the high-resolution spectra of C1s and Si_2_p cores are fitted to provide a basis for further study of the chemical structure of the *f*-SiO_2_ surface. Figure 2c,d shows the C1s XPS spectra of SiO_2_ and *f*-SiO_2_, respectively. Significantly, from the comparison of C1s spectra of SiO_2_, the C1s spectra of *f*-SiO_2_ peak at 284.8 eV caused by Cg sp^2^ and Cd sp^3^ evidently increase [26,27]. For SiO_2_ composed of Si-O-Si units, it is impossible for the carbon element to appear. However, the peaks caused by the existence of carbon also appear in C1s spectra of SiO_2_, which is caused by the heterocarbon introduced by the test conditions. The heterocarbon in test environment leads to the existence of C-C and C=O, resulting the appearance of binding energy peaks at 284.8 and 287.8 eV, respectively. However, under the same test environment, the peak strength at 284.8 eV of *f*-SiO_2_ is obviously stronger than that of SiO_2_, which indicates that the *f*-SiO_2_ particles possess more carbon. Thus, it is not difficult to understand that the increased binding energy peak at 284.8 eV is caused by the existence of Si-CH=CH_2_ and Si-CH_3_ on the surface of *f*-SiO_2_. The Si_2_p XPS spectra of SiO_2_ and *f*-SiO_2_ are shown in Figure 2e,f, respectively. The most typical difference is that the Si_2_p XPS spectra of *f*-SiO_2_ has a peak at 101.5 eV caused by the Si-C bonds and that no peak appears for SiO_2_, which further confirms the successful surface modification.

### 3.2. Particle Size and BET Analysis of f-SiO_2_

The particle size and specific surface area of silica have great influence on the reinforcement effect for liquid silicone rubber [28]. In general, particles with a greater specific surface area have a higher specific surface energy, and a system with a high surface energy is unstable, as the particles have a tendency to agglomerate in order to lower the specific surface energy. Hence, particles with higher specific surface area are inferior in dispersibility. The particle size distribution can reflect the aggregation state of nanoparticles to a great extent, and the specific surface is one of the key parameters of mechanical reinforcement for rubber materials. Therefore, it is necessary to study these two parameters of *f*-SiO_2_.

The particle size distributions of *f*-SiO_2_ are measured using the dynamic light scattering method, and the results are shown in Figure 3a. It is clearly observed that both the SiO_2_ and *f*-SiO_2_ particle size showed multiscale dispersion and that the maximum particle size reaches the micron scale. The particle sizes for SiO_2_ at maximum peaks are about 5872, 856, and 268 nm, respectively. After modifying, the particle sizes at maximum peak shifted to 5231, 624, and 67 nm, respectively. The SEM images also intuitively proved that the agglomeration degree of the SiO_2_ particles decreased after the modification process, as depicted in Figure 3c,d. The reduction in the degree of agglomeration for SiO_2_ is likely attributed to the improvement of hydrophobicity. It is well known that the large amount of hydroxyl exists on the surface of SiO_2_ and that hydrogen bonding between hydroxyl groups occurs easily, resulting the occurrence of agglomeration. After modification with TMDVS, the hydrophobicity of SiO_2_ is greatly improved, which weakens the hydrogen bonding force between SiO_2_ particles and reduces the agglomeration degree. The hydrophobicity of SiO_2_ before and after modification is shown in Figure 3d.

The effects of the amount of TMDVS on the BET surface area of *f*-SiO_2_ particles are shown in Table 1. It can be seen that the mass fraction of TMDVS relative to SiO_2_ has an obvious effect on BET surface area. The BET specific surface area of unmodified SiO_2_ particles is the highest, 258.65 m^2^/g, while the specific surface areas of modified SiO_2_ particles decrease to a certain extent with the increase in TMDVS content. Among them, the TMDVS with a mass fraction of 20 wt.% relative to SiO_2_ was remarkably efficient, and the BET surface area of *f*-SiO_2_ particles was decreased by 25.47% relative to SiO_2_. This is closely related to the change in physicochemical state of the surface and micropores of SiO_2_. [14,29,30] When the SiO_2_ particles are modified by TMDVS, both the inner surfaces of the particles and the micropores are occupied by TMDVS, resulting in a decrease in the BET surface area of SiO_2_ particles. However, it can be seen that after the amount of TMDVS reaches 40%, the value of BET surface area has a little increase with the increasing amount of TMDVS, which is closely related to the dispersion state of the modified SiO_2_ particles. The reason can be explained as follows: The increasing TMDVS content improves the dispersion level of SiO_2_ particles, which can be verified via the SEM micrographs in Figure 3b,c. The improvement of dispersion level means that there are more micropores in SiO_2_ particles, thus contributing to a slight increase in the BET surface area of SiO_2_ particles.

### 3.3. TG-DSC Analysis of f-SiO_2_

Figure 4 shows the TG–DSC curves of SiO_2_ and *f*-SiO_2_, respectively. From the TG curves, it can be seen that both the SiO_2_ and *f*-SiO_2_ exhibit typical two-stage thermal decomposition. The first-stage weight losses (<150 °C) for SiO_2_ and *f*-SiO_2_ are about 2.68 wt.% and 2.91 wt.%, respectively. Though the samples are dried at 100 ◦C for 24 h to remove the redundant water on the surface, the obvious weight loss in samples can be observed in the curves. This phenomenon is attributed to the removal of the bound water absorbed on the surface of SiO_2_ and *f*-SiO_2_. The second-stage weight losses (200–700 °C) for SiO_2_ and *f*-SiO_2_ are about 1.47 wt.% and 4.31 wt.%, respectively. For SiO_2_, the weight loss is mainly derived from the dehydration condensation of Si-OH in SiO_2_ particles [14]. However, the weight loss of *f*-SiO_2_ is higher than that of SiO_2_ in the curves. This indicates not only a dehydration reaction of Si-OH between SiO_2_ particles but also a thermal decomposition reaction caused by organic components occurring on the surface of SiO_2_ particles after modification, which can also be confirmed by the DSC result. From the DSC curves, it can be seen that the *f*-SiO_2_ shows an obvious exothermic peak around 575 °C, while the SiO_2_ shows no exothermic peak in the same temperature range. Moreover, the grafting ratio of the *f*-SiO_2_ was calculated from Equation (1) as 2.84%.

### 3.4. Rheological Properties

The effects of *f*-SiO_2_ and SiO_2_ additions on the rheological properties of SR are shown in Figure 5. As depicted in Figure 5a, all of the uncured composites exhibit the typical shear-thinning behavior of non-Newtonian fluids. At lower shear rates, the pseudo-cross-linking phenomenon caused by the van der Waals force between SiO_2_ particles and SR molecular chains makes the composites present a higher viscosity. With the increase in shear rate, the pseudo-crosslinking structure is gradually destroyed and the SR molecular chains relax, which causes the viscosity of the composites to decrease. Additionally, the difference in viscosity between the uncured *f*-SiO_2_/SR (u*f*-SiO_2_) and SiO_2_/SR composites (uSiO_2_) enlarges with an increase in filler content. In the whole shear rate range, the viscosity of u*f*-SiO_2_ is always lower than that of uSiO_2_, which is attributed to the better compatibility between *f*-SiO_2_ particles and SR compared to SiO_2_. Typically, When the *f*-SiO_2_ content reaches 10 wt.%, the viscosity of u*f*-SiO_2_R is only 29.6 Pa·s at the shear rate of 600 s^−1^, while the viscosity of uSiO_2_ reaches an amazing 48.3 Pa·s at the same filler content.

In order to quantitatively compare the effects of *f*-SiO_2_ and SiO_2_ on the viscosity of its composites, the power law model (Equations (4) and (5)) is used [14,31,32]. The fitting curves of shear stress and shear rate of the two kinds of composites are shown in Figure 5b, and the parameters of the fitting curves are listed in Table 2.
(4)$$\sigma =k^{\prime }\cdot \gamma^{n}$$
(5)η=σ/γ
where *σ*, *k*′, *γ*, *n*, and *η* are shear stress, consistency coefficient, shear rate, rheological index, and correlating viscosity, respectively. Among them, the rheological index *n* reflects the sensitivity of the fluid to shear, while the consistency coefficient *k*′ directly reflects the viscosity of the fluid. It can be seen that the value of *n* for u*f*-SiO_2_ is always higher than that for uSiO_2_ at the same filler content, which indicates that the u*f*-SiO_2_ possess stronger shear sensitivity [33]. Moreover, it is worth noting that the values of *k*′ for the *f*-SiO_2_/SR composites system is much smaller than that of uSiO_2_ overall as filler content increases, suggesting the lower viscosity and better manufacturability for u*f*-SiO_2_.

### 3.5. Micrographs Analysis

Figure 6 shows the TEM morphologies of SiO_2_ and *f*-SiO_2_ at 10 wt.% content in SR matrix. As shown in Figure 6a, the SiO_2_ particles cannot be evenly dispersed into the SR matrix, and a large number of aggregates appear, which is closely related to the compatibility between the particles and SR matrix. The surface of SiO_2_ has a large amount of hydrophilic Si-OH groups, which are incompatible with the hydrophobic SR matrix, resulting in the poor dispersion of SiO_2_ in SR [21]. The cluster area as shown in Figure 6b can be observed using a high-magnification TEM. It can be clearly observed that some aggregation of SiO_2_ particles exists in SR matrix. The aggregation degree between the *f*-SiO_2_ and SR matrix is lower than that of SiO_2_; only a relatively small cluster *f*-SiO_2_ particles is formed at about 85 nm in size, as shown in Figure 6c,d, which is due to the incomplete substitution of hydrophilic Si-OH groups on the surface of *f*-SiO_2_ particles. These results indicate that modified process of SiO_2_ has a good effect on the dispersion of *f*-SiO_2_ in SR matrix because the hydrophilic surface of unmodified SiO_2_ particles changes to the SR-friendly hydrophobic state.

### 3.6. Thermal Stability Analysis of SiO_2_/SR Composites

Figure 7 shows the TG curves of the SR composites at 6 wt.% and 10 wt.% filler contents. The 10% decomposition temperature (*T*_10_), the maximum thermal decomposition temperature (*T_max_*), and 1000 °C residual (*R*_1000_) of the *f*-SiO_2_/SR composites obtained from TG curve are listed in Table 3. For SR, the temperatures of *T*_10_, *T_max_*, and *R*_1000_ were 483.4 °C, 554.3 °C, and 58.6%, respectively. After being reinforced by SiO_2_ and *f*-SiO_2_, the *T*_10_, *T_max_*, and *R*_1000_ of all composites show different degrees of enhancement, which indicates that the characteristics of nanostructures and high specific surface area for SiO_2_ particles can effectively improve the thermal stability of SR. In particular, at the same loading of SiO_2_, the *f*-SiO_2_/SR composites show better thermal stability than SiO_2_/SR composites. The *T_10_*, *T_max_*, and *R*_1000_ of 10 wt.% SiO_2_/SR composites are 507.9 °C, 593.5 °C, and 77.2%, respectively, while the 10 wt.% *f*-SiO_2_/SR composites system can reach 518.3 °C, 603.2 °C, and 78.9%, respectively, which is caused by the following reasons: (1) After modification, the *f*-SiO_2_ still maintains a relatively large specific surface area, which is beneficial for *f*-SiO_2_ to restrict the molecular chains motion of SR. (2) The *f*-SiO_2_ has better dispersibility than SiO_2_ in SR matrix, which means that the number of particles per unit volume increases. Thus, more SR molecular chains are restricted, contributing to the improvement of thermal stability [28]. (3) The strong chemical bonds between the interface of *f*-SiO_2_ and SR require more energy to destroy, resulting in the better thermal stability compared to SiO_2_/SR composites combined by hydrogen bonds.

### 3.7. Thermal Conductive Property

Liquid silicone rubber is widely used as the matrix resin of high temperature thermal insulation materials because of its excellent heat resistance. For polymer-based thermal insulation materials, the polymer matrix acting as continuous phase of heat conduction has a great influence on the thermal conductivity of its composites. For example, in our previous work, the thermal conductivity of the silicone rubber as the matrix is 0.185 W/(mK), while that of the hollow silica microspheres (HSM) as thermal insulation functional phase is only 0.012 W/(mK), which is one order of magnitude lower than that of the matrix [34]. Therefore, it is necessary to study the variation of thermal conductivity of SR with the change of SiO_2_ particles content, which may provide some help in the design of SR-based thermal insulation materials.

The thermal conductivities of *f*-SiO_2_/SR and SiO_2_/SR composites are shown in Figure 8. The thermal conductivity of pure SR is only 0.121 W/(mK). After adding the *f*-SiO_2_ and SiO_2_, the thermal conductivities of both the two kinds of composites are improved. Overall, the thermal conductivity of *f*-SiO_2_/SR composites is always higher than that of SiO_2_/SR composites system at the same filler content. After adding 10 wt.% filler contents, the thermal conductivities of *f*-SiO_2_/SR and SiO_2_/SR composites reach 0.161 W/(mK) and 0.143 W/(mK), respectively, which are 41.3% and 26.4%, respectively, higher than that of SR. In particular, it should be noted that the thermal conductivity is not too different when the filler content is less than 6 wt.%. With the increase in filler content, the difference in thermal conductivity between the two types of composites increases obviously. This phenomenon may be explained by the following reasons: (1) the thermal conductivity of silica is generally considered to be 1.4 W/(mK), which is higher than that of SR in this study. Therefore, the *f*-SiO_2_ and SiO_2_ in its composites can be regarded as heat-conducting particles. The thermal conductivity of composites depends on the matrix polymer, fillers, and the combination state between them [35,36,37]. When the filler content is low, the fillers can be evenly dispersed in SR, but there is no contact and interaction between the fillers, which leads to the little contribution to the improvement of thermal conductivity of the whole composite system. With the increase in the filler content, the number of fillers reaches the critical point and the fillers begin to contact each other, forming a thermal conductivity network, which causes the thermal conductivity of the composites to begin to show an obvious upward trend. (2) Heat transfer between conductive particles and polymer matrix is realized via phonon propagation [24]. The interface state between *f*-SiO_2_ and SiO_2_ with SR is different. The interface between *f*-SiO_2_ and SR is connected by chemical bonds, while the interface between SiO_2_ and SR is only connected by hydrogen bonds. As reported in related literature, after surface modification, the bonding force between the conductive particles and the polymer matrix can be greatly increased because of the decrease of interface thermal resistance, contributing to the enhancement of phonon propagation and the thermal conductivity [24,38].

In order to further study the influence of surface modification on the thermal conductivity of the composites, the thermal conductivity of the *f*-SiO_2_/SR and SiO_2_/SR composites are analyzed by using the Maxwell–Eucken model [39]. The influence of the interface thermal resistance and the filler shapes on the thermal conductivity of the composite is considered in this model, as described by Equation (6).
(6)$$k_{c}=k_{p}\frac{k_{f}\left[1+\left(\psi -1\right)\alpha \right]+\left(\psi -1\right)k_{p}+\left(\psi -1\right)\varphi_{f}\left[k_{f}\left(1-\alpha \right)-k_{p}\right]}{k_{f}\left[1+\left(\psi -1\right)\alpha \right]+\left(\psi -1\right)k_{p}-\varphi_{f}\left[k_{f}\left(1-\alpha \right)-k_{p}\right]}$$
where *φ_f_*, *k_c_*, *k_p_*, and *k_f_* are the volume fraction of conductive fillers and the thermal conductivities of the composites, SR matrix, and fillers, respectively. *ψ* represents the shape factor of conductive fillers, which is related to filler sphericity. The closer the value of *ψ* is to 3, the more spherical the fillers are. *α* is the interfacial thermal resistance factor, which is an evaluation of interfacial thermal resistance. The larger the value of *α* is, the larger the thermal resistance between fillers and polymer matrix is. In this study, both the values of *ψ* and *α* are unknown. Therefore, it can be first assumed that no interfacial thermal resistance exists between the fillers and SR matrix, that is, α = 0. In this case, Equation (3) is used to generate a series of derived curves (Figure 9a,b) between the thermal conductivity of the composites and the filler content at different filler shape factors (*ψ*). In order to distinguish the *ψ* value of *f*-SiO_2_/SR and SiO_2_/SR composites, the *ψ* values of *f*-SiO_2_/SR composites and SiO_2_/SR composites are named as *ψ*_1_ and *ψ*_2_, respectively. By comparing with the experimental data of the composites, it can be concluded that the shape factors are in good agreement with the experimental data when the value of *ψ*_1_ and *ψ*_2_ are 6 and 4, respectively. The different shape factors of the two kinds of composite systems indicate that the surface modification process has a great influence on the shape of the fillers, which is consistent with the result of SEM in Figure 3b,c. However, considering the existence of interfacial thermal resistance between the fillers and SR matrix in practice, it can be inferred that the actual values of *ψ* are larger than the calculated values. Therefore, for further investigation, the values of *ψ*_1_ and *ψ*_2_ are chosen to be 7 and 5, respectively. Based on this assumption, the relationships between the thermal conductivity and filler content of the composites under different interfacial thermal resistance factors (*α*) are shown in Figure 9c,d. It can be seen that the experimental data of the composites match the corresponding predicted curves when the interfacial thermal resistance factors of *f*-SiO_2_/SR and SiO_2_/SR composites are about 0.1 and 0.2, respectively. This result indicates that the interfacial thermal resistance between the *f*-SiO_2_ and SR matrix is lower than that of SiO_2_ and SR matrix, attributing to the enhancement of phonon propagation after the chemical bonding between *f*-SiO_2_ and SR matrix. In particular, when the fillers content is exceeds 6 wt.%, the experimental data deviate greatly from the corresponding curves. This is because that the Maxwell–Eucken model is suitable for the thermal conductivity of composites with low-volume-fraction fillers [24]. Therefore, the better dispersion and the lower interfacial thermal resistance with SR matrix are the main reasons for the higher thermal conductivities of *f*-SiO_2_/SR composites compared to SiO_2_/SR composites.

### 3.8. Mechanical Property

The mechanical properties of pure SR and its composites with different contents of SiO_2_ and *f*-SiO_2_ are measured. Figure 10a,b shows the typical stress–strain curves of SiO_2_/SR and *f*-SiO_2_/SR composites, respectively. Evidently, it can be seen that the mechanical properties of both SiO_2_/SR and *f*-SiO_2_/SR composites increase with additions of corresponding particles. The ultimate tensile stress (tensile strength) for the composites is extracted from the curves just after failure, and the corresponding comparison results are shown in Figure 10c. It can be seen that there are obvious differences between SiO_2_ and *f*-SiO_2_ in improving the mechanical properties of SR. The tensile strength of SiO_2_/SR composites increases from 3.42 MPa to 4.83 MPa when the filling amount increases from 0 to 10 wt.%, about a 50.1% increase. In contrast, the tensile strength of *f*-SiO_2_/SR composites increases to 5.86 MPa, a 74.3% enhancement of SR at 10 wt.% filler content. Therefore, it can be concluded that the *f*-SiO_2_ has a better mechanical reinforcement effect on SR than SiO_2_ does.

The reinforcing effect of nanomaterials on rubber is closely related to the dispersion state of nanomaterials in the matrix and the interface force [40,41]. In this study, the dispersion state of SiO_2_ in SR matrix will directly affect the mechanical strengthening effect. As shown in Figure 6, it can be intuitively seen that the *f*-SiO_2_ particles distribute more uniformly in SR matrix than SiO_2_ particles do even though agglomerations are not avoided. However, the size of agglomerations in SiO_2_/SR composites include several microns, submicrons, and even nanometers. Considering such a size distribution, hierarchical network structure can be consequently formed between the different-sized aggregates and SR molecular chains, and the stress transfer ability of such hierarchical polymer/filler networks is considered as the main reason for the improved strength of the composites [42]. Therefore, it is not hard to understand that the *f*-SiO_2_/SR composites system with better dispersion in SR matrix will have a more perfect hierarchical network structure, contributing to the better mechanical properties compared to the SiO_2_/SR composites system. In addition, the interface force between SiO_2_ particles and SR matrix is also a key factor affecting the mechanical properties for the composites, and the interface force between aggregates and matrix is beneficial for improving the overall strength [43]. For SiO_2_/SR composites, the interface force between SiO_2_ particles and SR matrix is mainly the van der Waals force, which is a type of relatively weak interface force. Thus, during the stretching process, the SiO_2_ particles are easily deboned between the interfaces of the composites, leading to the failure of tensile strength [42]. For *f*-SiO_2_/SR composites, the Si-CH=CH_2_ groups on the surface of *f*-SiO_2_ can polymerize with the Si-H groups in SR matrix in situ, and thus, the interface force between SiO_2_ particles and the SR matrix is mainly the chemical bond force, which is a type of strong interface force. The comparison results of elongation at break for the composites are shown in Figure 10d. With the increase in filler content, the elongation at break for both the SiO_2_/SR and *f*-SiO_2_/SR composites exhibits a continuous increasing trend. However, the growth of *f*-SiO_2_/SR composites is lower than that of SiO_2_/SR composites. The elongation at break of SiO_2_/SR composites increases from 115.1% to 146.2%, and the elongation at break increases by 27.2% when the filling amount increases from 0 to 10 wt.%. In contrast, the elongation at break of *f*-SiO_2_/SR composites increases to 137.3%, only 19.3% higher than that of pure SR at 10 wt.% filler loading, which is caused by the stronger binding effect of *f*-SiO_2_ on SR molecular chains compared to that of SiO_2_. As is known, for rubber materials, the rubber molecular chain is very long, and there are physical or chemical interactions between the molecules, forming a large network. In addition, the rubber molecular chain has good flexibility and a low interaction force. When large deformation occurs, the molecular chain network will deform, and when the external force is released, the networks will recover. This is why rubber materials have good elasticity. For *f*-SiO_2_/SR composites, the motion of the SR molecular chain is greatly limited because of the strong chemical bonds between *f*-SiO_2_ particles and the SR matrix. For SiO_2_/SR composites, due to the relatively weak van der Waals force between SiO_2_ particles and the SR matrix, the movement of the SR molecular chain will not be greatly affected, resulting in better extensibility compared to the *f*-SiO_2_/SR composites system.

As a typical cushioning material, the compression properties and hardness of silicone rubber materials are very important for practical application. Figure 11 shows the compression property and hardness variation law with the increase of filler content. As shown in Figure 11a,c, with the increase in filler content, the compressive performances of both the *f*-SiO_2_/SR and SiO_2_/SR composites show a continuous increasing trend. As shown in Figure 11b,d. It can be seen that the *f*-SiO_2_ can effectively improve the compressive modulus of SR. With the filling amount increasing from 0 to 10 wt.%, the compressive modulus of SR increases from 4.48 MPa to 9.82 MPa. For SiO_2_/SR composites, the compressive modulus only increases from 4.48 MPa to 8.32 MPa. The difference of reinforcement to SR between SiO_2_ and *f*-SiO_2_ may be due to the dispersion of fillers and the strength of their interaction with SR molecular chains. The compression failure mechanism of composite materials is generally considered to be the “Rosen model”. In this model, tensile and shear modes are the main modes of composites failure caused by compressive load [44]. The better dispersion of *f*-SiO_2_ is more beneficial to the improvement of the tensile strength of SR than to that of SiO_2_, which is consistent with the result of Figure 10. Additionally, as reported in the literature [45], the mechanical properties of surface-modified carbon fibers (CFs) and polypropylene (PP) were studied. The results showed that the shear modulus of the composites with modified CFs showed better performance than that of unmodified CFs, which is attributed to the stronger force of the modified CFs with PP molecular chains. Therefore, it is not difficult to understand that the *f*-SiO_2_/SR composites system will have the better compression performance than that of SiO_2_/SR composites system because the *f*-SiO_2_ has stronger interface force with SR molecular chains than that of SiO_2_.

The hardness of the SiO_2_/SR and *f*-SiO_2_/SR composites are also shown in Figure 11b,d, respectively. It can be seen that the hardness of both two kinds of composites increase with the increase of fillers content. The maximum value of hardness for *f*-SiO_2_/SR composites can reach 43°, while the value of SiO_2_/SR composites is only 37°. The improvement of hardness of the composites is mainly caused by the restriction of the rigidity SiO_2_ particles to the flexible SR molecular chains [46]. For *f*-SiO_2_/SR composites, the Si-CH=CH_2_ on the surface of *f*-SiO_2_ may cause an increase in crosslinking density, resulting in the higher hardness of the composites compared to SiO_2_/SR composites. In order to further confirm this hypothesis, the crosslinking densities of SiO_2_/SR and *f*-SiO_2_/SR composites with different filler content are tested and the relevant results are shown in Table 4. Combined with the results of the above mechanical properties, it can be seen that the SiO_2_ particles tends to improve the crosslinking density alongside reinforcing the networks, and this result is also consistent with the relevant literature reports [47,48]. The increase of crosslinking density with the increase of filler content in this study is closely related the large specific surface area of SiO_2_ particles. The physical adsorption of SiO_2_ particles can greatly increase the entanglement probability between SR molecular chains and rigid SiO_2_ particles, which helps to improve the crosslinking density of the composites. the *f*-SiO_2_/SR composites system, there are not only physical entanglements but also chemical bonding through in situ polymerization between SR molecular chains and *f*-SiO_2_ particles, which may further improve the crosslinking density of the composites.

## 4. Conclusions

A type of vinyl-modified SiO_2_ particle (*f*-SiO_2_) was prepared through a surface modification of hydrophilic silica with the tetramethyldivinyldisilazane (TMDVS). The chemical structures of *f*-SiO_2_ were characterized by FT-IR and XPS, and the results confirmed the existence of typical characteristic peak caused by Si-CH=CH_2_ groups, suggesting successful modification. The effect of modification on particle size distribution and specific surface area were also studied, the results of which demonstrated that the aggregation of hydrophobic particles is greatly reduced. TEM observations indicated that the *f*-SiO_2_ particles were fairly well dispersed in SR composites in the form of clusters 85 nm in size at maximum. The rheological results show that the *f*-SiO_2_/SR composites system possessed lower viscosity and better manufacturability than the SiO_2_/SR composite. The viscosity of *f*-SiO_2_/SR composite was only 29.6 Pa·s at the shear rate of 600 s^−1^, while the viscosity of SiO_2_/SR composites system reached an amazing 48.3 Pa·s at the same filler content. The TG results indicated that the thermal stability of *f*-SiO_2_/SR composites was better than that of SiO_2_/SR composites at same filler content, which is attributed to to the large specific surface area that remained and good dispersibility in SR matrix. The thermal conductivity results indicated that the *f*-SiO_2_ is not conducive to the preparation of low-thermal-conductivity SR matrix because of the high dispersion and low interfacial thermal resistance with matrix, especially at high filler content. The results of mechanical properties showed that the *f*-SiO_2_ had a good reinforcement effect on SR matrix and that the maximum tensile strength of *f*-SiO_2_/SR composite could reach 5.86 MPa, while the SiO_2_/SR composite is only 4.83 MPa at 10 wt.% filler contents. Additionally, the compressive strength had also been improved; the maximum compressive module of *f*-SiO_2_/SR composite and SiO_2_/SR composite increased from 4.48 MPa to 9.82 MPa and 8.32, respectively. These improvements in tensile and compressive strength were attributed to the covalent bonding between the Si-CH=CH_2_ groups on the surfaces of the *f*-SiO_2_ particles and the Si-H groups in SR.

## Figures and Tables

**Figure 1 polymers-15-01224-f001:**
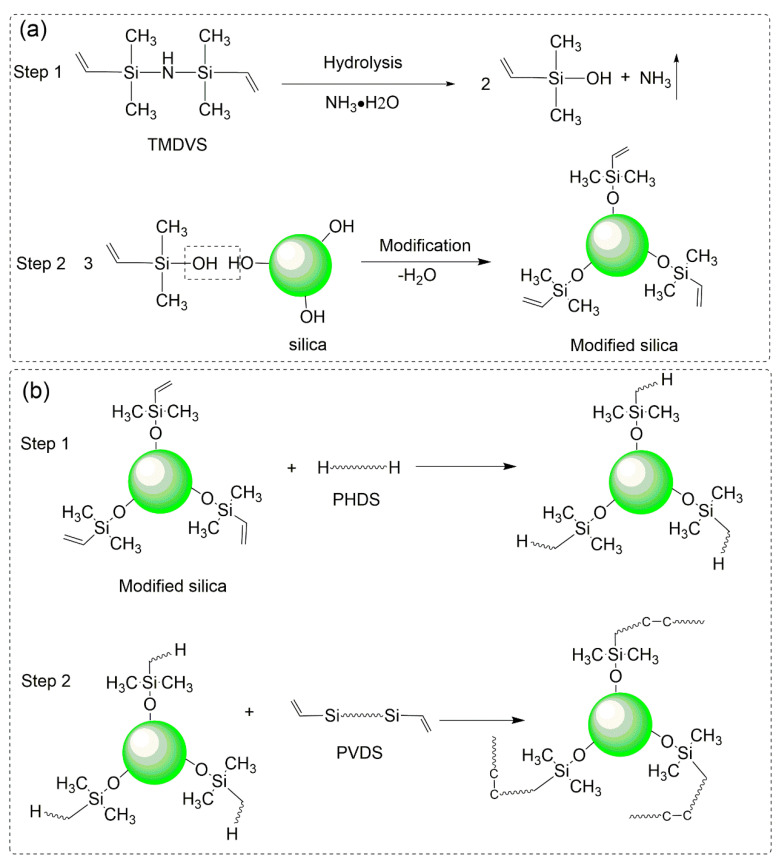
Schematic diagram of (**a**) TMDVS modification process on silica surface; (**b**) mechanism of in situ polymerization between *f*-SiO_2_ and SR matrix.

**Figure 2 polymers-15-01224-f002:**
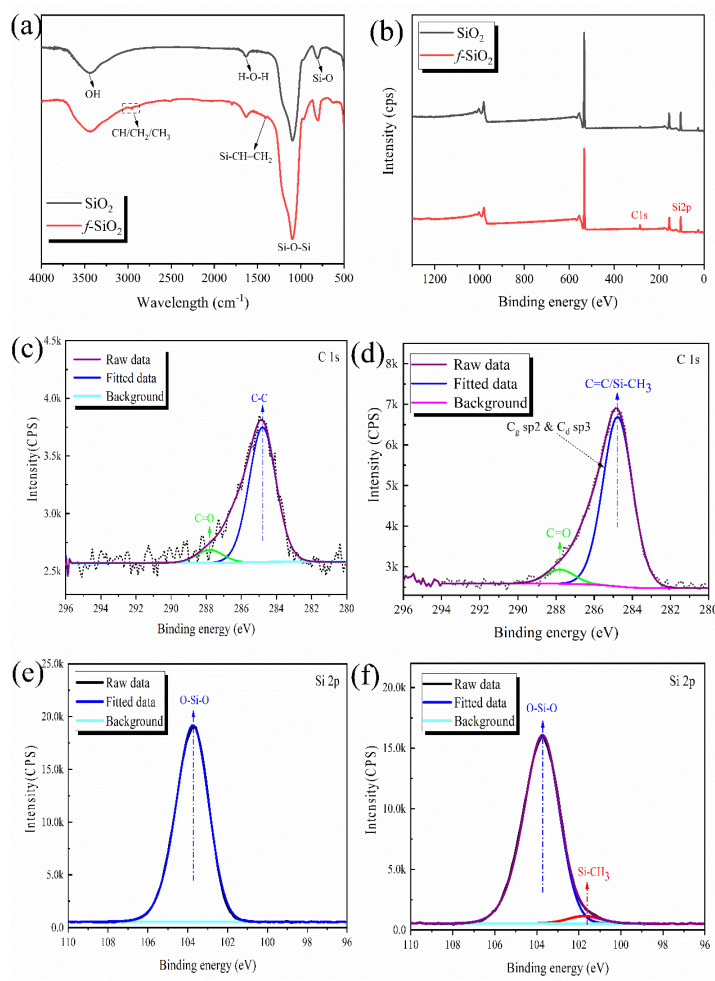
(**a**) FT-IR spectra and (**b**) wide-scan XPS spectra of SiO_2_ particles; high-resolution XPS spectra: (**c**,**d**) C1s peaks of *f*-SiO_2_ and SiO_2_, respectively; (**e**,**f**) Si_2_p peaks of *f*-SiO_2_ and SiO_2_, respectively.

**Figure 3 polymers-15-01224-f003:**
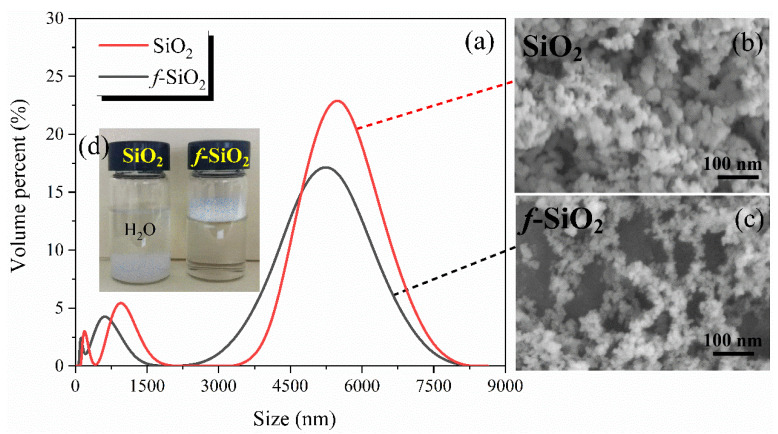
(**a**) Particle size distribution; (**b**,**c**) SEM images of SiO_2_ before and after modification; (**d**) hydrophobicity of SiO_2_ and *f*-SiO_2_.

**Figure 4 polymers-15-01224-f004:**
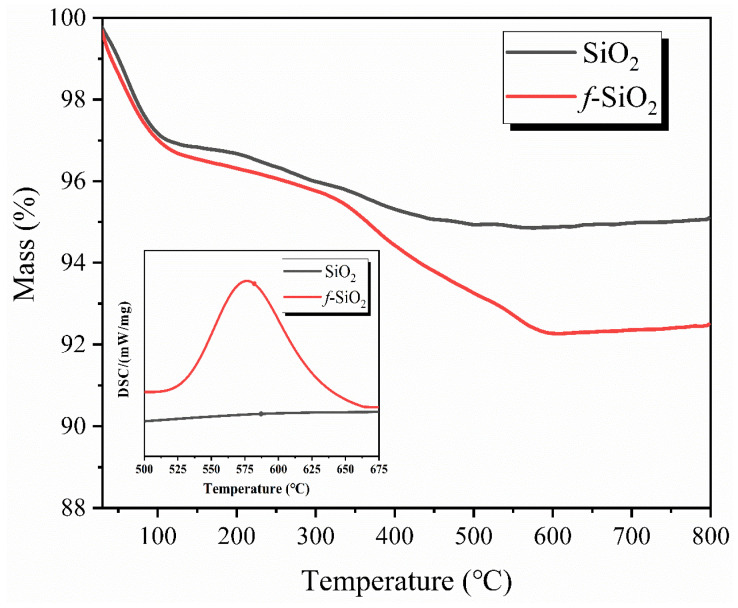
TG–DSC curves of SiO_2_ and *f*-SiO_2_, respectively.

**Figure 5 polymers-15-01224-f005:**
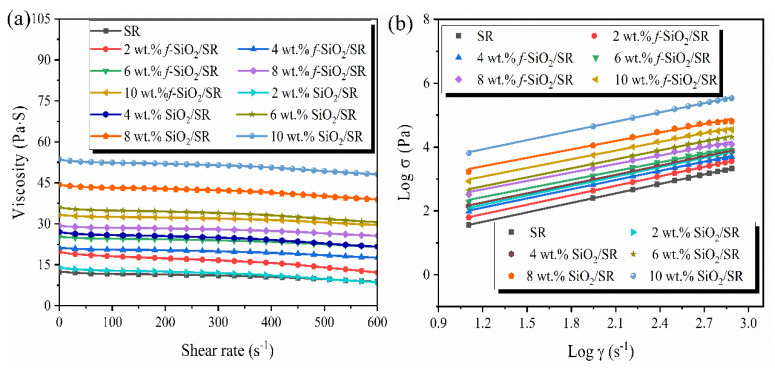
(**a**) Shear viscosity of SR and its composites filled with *f*-SiO_2_ and SiO_2_; (**b**) relationship between apparent shear rate and shear stress of SR and its composites filled with *f*-SiO_2_ and SiO_2_.

**Figure 6 polymers-15-01224-f006:**
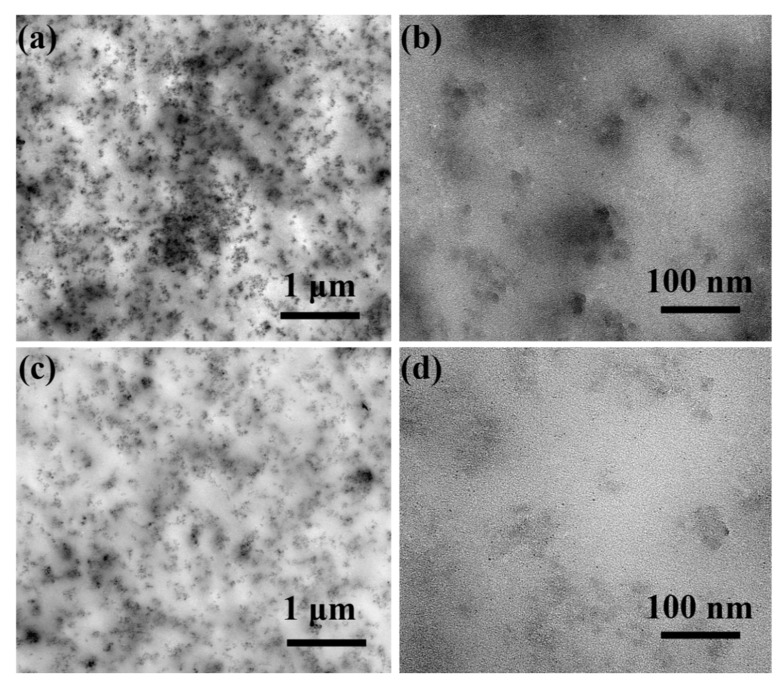
TEM morphologies of (**a**,**b**) the SiO_2_/SR and (**c**,**d**) the *f*-SiO_2_/SR composites at 10 wt.% filler content.

**Figure 7 polymers-15-01224-f007:**
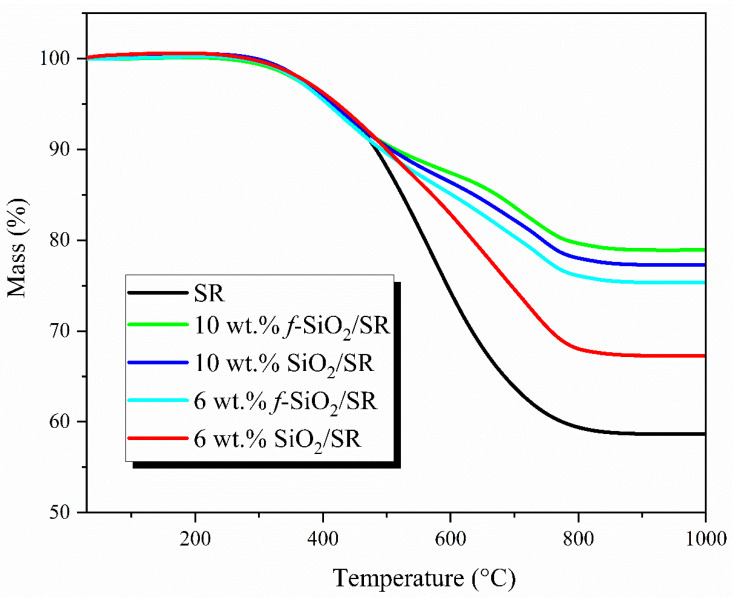
Thermal stability of the SR composites filled with *f*-SiO_2_ and SiO_2_ at the filler content of 6 wt.% and 10 wt.%.

**Figure 8 polymers-15-01224-f008:**
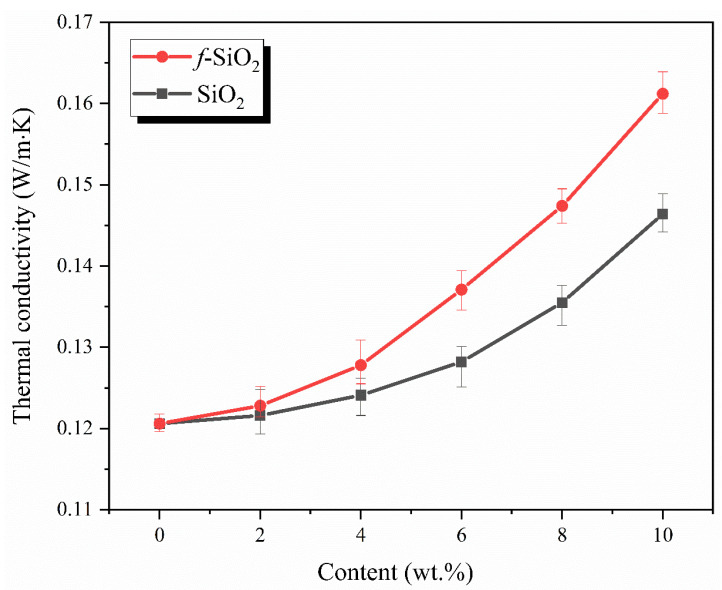
Thermal conductivity of *f*-SiO_2_/SR and SiO_2_/SR composites with different fillers loadings.

**Figure 9 polymers-15-01224-f009:**
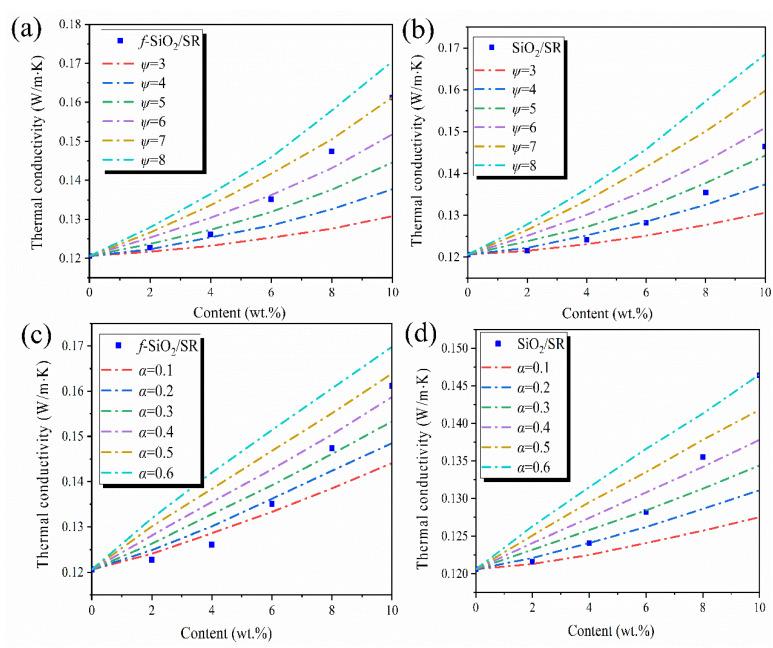
(**a**,**b**) The effect of filler shapes on thermal conductivity of composites and (**c**,**d**) the effect of interfacial thermal resistance on thermal conductivity of composites.

**Figure 10 polymers-15-01224-f010:**
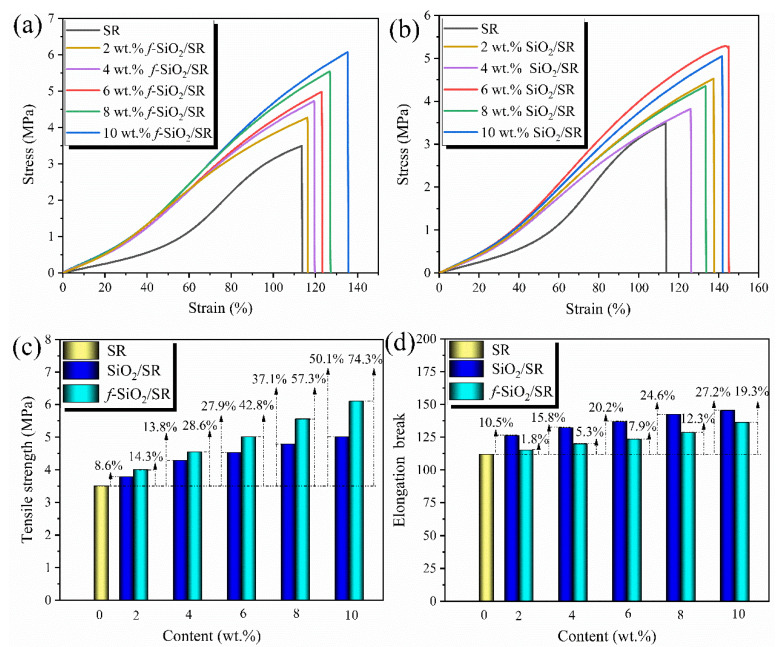
(**a**,**b**) Tensile stress–strain curves of *f*-SiO_2_/SR and SiO_2_/SR composites and (**c**) tensile strength of SR and its composites filled with *f*-SiO_2_ and SiO_2_; (**d**) elongation break of SR and its composites filled with *f*-SiO_2_ and SiO_2_.

**Figure 11 polymers-15-01224-f011:**
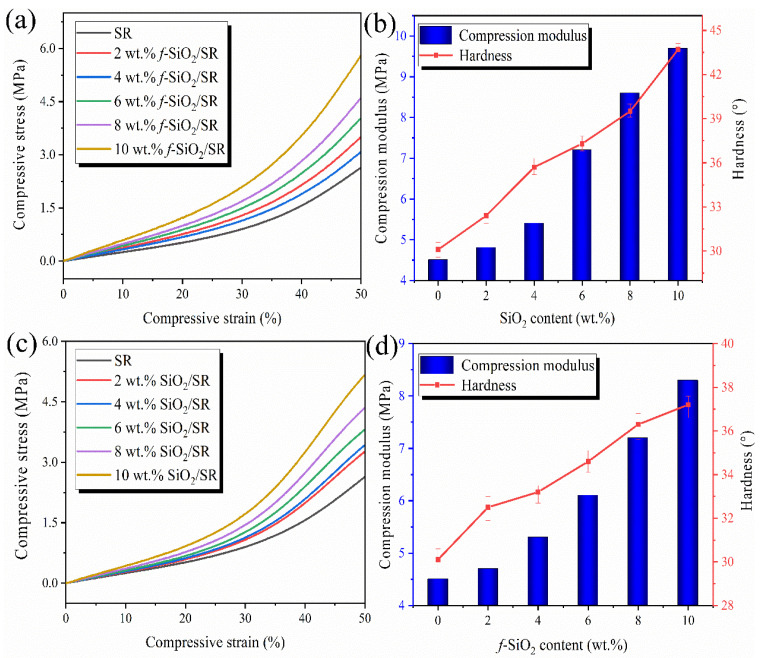
(**a**,**c**) Compressive stress–strain curves of *f*-SiO_2_/SR and SiO_2_/SR composites; (**b**,**d**) compressive modulus and hardness of *f*-SiO_2_/SR and SiO_2_/SR composites.

**Table 1 polymers-15-01224-t001:** The effect of TMDVS content on the BET surface area of *f*-SiO_2_.

BET Surface Area (m^2^/g)	Mass Fraction of TMDVS Relative to SiO_2_ (wt.%)					
	0	20	40	60	80	100
*f*-SiO_2_	258.65	192.76	213.13	218.27	221.46	227.29

**Table 2 polymers-15-01224-t002:** The power law parameters of *f*-SiO_2_/SR and SiO_2_/SR composites.

Samples	*n*	$k^{\prime }$	Samples	*n*	$k^{\prime }$
SR	0.978	2.017	/	/	/
2 wt.% *f*-SiO_2_/SR	0.975	2.841	2 wt.% SiO_2_/SR	0.967	3.887
4 wt.% *f*-SiO_2_/SR	0.953	8.763	4 wt.% SiO_2_/SR	0.921	12.365
6 wt.% *f*-SiO_2_/SR	0.924	20.136	6 wt.% SiO_2_/SR	0.887	28.760
8 wt.% *f*-SiO_2_/SR	0.887	38.456	8 wt.% SiO_2_/SR	0.842	43.566
10 wt.% *f*-SiO_2_/SR	0.853	49.453	10 wt.% SiO_2_/SR	0.803	61.391

**Table 3 polymers-15-01224-t003:** Thermal stability of *f*-SiO_2_/SR composites.

Samples	*T*_10_ (°C)	*T_max_* (°C)	*R*_1000_ (%)
SR	483.4	554.3	58.6
10 wt.% *f*-SiO_2_/SR	518.3	603.2	78.9
10 wt.% SiO_2_/SR	507.9	593.5	77.2
6 wt.% *f*-SiO_2_/SR	492.1	587.7	75.3
6 wt.% SiO_2_/SR	500.3	571.3	67.2

**Table 4 polymers-15-01224-t004:** Comparison of *M_C_* for SiO_2_/SR and f-SiO_2_/SR composites.

Samples	*M_C_* (g/mol)					
	0	2 wt.%	4 wt.%	6 wt.%	8 wt.%	10 wt.%
*f*-SiO_2_/SR	2159	2278	2267	2372	2451	2512
SiO_2_/SR	2159	2367	2416	2563	2598	2725

## Data Availability

Not applicable.

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
