# Peer review of "Effects of Vinyl Functionalized Silica Particles on Thermal and Mechanical Properties of Liquid Silicone Rubber Nanocomposites"

_polymers, 2023, doi:10.3390/polym15051224_

Round 1

Reviewer 1 Report

The paper entitled "Effects of vinyl functionalized silica particles on thermal and mechanical properties of liquid silicone rubber" by Q. Zhou reports on the functionalization of the silica particles with vinyl groups in order to improve their performance as fillers in liquid silicone rubber and its nanocomposite films.

The idea of this paper is already known and applied in numerous studies, so that I consider that overall paper lacks originality. The authors describe also some nanocomposites based on functionalized silica particles, so that the title is not the most suggestive.

The introduction Section must be improved, these are just general reports on the silica particles, the idea of the authors does not come to bring a new strategy, a new reaction, or something new etc.

Fig. 1 is not suggestive. The same composites were prepared for non-functionalized silica?  How the authors prepared the other composites?

In the Results and discussion section there are some errors in the FTIR interpretation in the attribution of the vinyl groups. The authors need to improve this  section.

In the particle size and BET analysis there are only a few differences between the SiO2 and f-SiO2 in therms of aggregation of the particles and their BET surface. Why the BET surface is important for the properties of the mixtures with SR and of composites? The dispersion of the particles in the in situ generated silicone matrix was not proved. Fig. 3 suggested only the hydrophilicity of the particles, the SEM images are not conclusive. SEM on the surface and in the section of the composites is needed.

The grafting ratios must be determined from TGA data. The DSC endotherm process at 575 oC is not conclusive, at that temperature there is a mass loss in the TGA curve. So that this section needs improvements.

Fig. 6 need some comments, the differences are not such conclusive.

The tensile-stress curves for nonfunctionalized SiO2 based composites indicate a higher mechanical performance than the composites based on f-SiO2.

Based on these observations I suggest major revision for this paper, especially on the novelty and the performance of these composites. 

Author Response

Reviewer #1:

  1. The paper entitled "Effects of vinyl functionalized silica particles on thermal and mechanical properties of liquid silicone rubber" by Q. Zhou reports on the functionalization of the silica particles with vinyl groups in order to improve their performance as fillers in liquid silicone rubber and its nanocomposite films. The idea of this paper is already known and applied in numerous studies, so that I consider that overall paper lacks originality. The authors describe also some nanocomposites based on functionalized silica particles, so that the title is not the most suggestive.

Response: Thanks for your suggestion. Your opinion is of great help to us. The idea of this paper is mainly the in-situ modification between the particles and SR matrix by modifying silica particles, thus enhancing the combination. Meanwhile, the effects of modifiers on the BET surface area, particle size distribution and rheological properties of the particles, which were lacking in previous in situ modifications, are systematically described. And, comparing the previous study, this paper reduces the content of the modified filler with enhanced thermal and mechanical properties. Finally, after careful consideration, we decided to revise the title of the article from "Effects of vinyl functionalized silica particles on thermal and mechanical properties of liquid silicone rubber" to “Effects of vinyl functionalized silica particles on thermal and mechanical properties of liquid silicone rubber nanocomposites”. It makes the title more comprehensive. This revise is marked in red in the manuscript.

  1. The introduction Section must be improved, these are just general reports on the silica particles, the idea of the authors does not come to bring a new strategy, a new reaction, or something new etc.

Response: Thanks for your suggestion. We have simplified the language and clarified the logic of the introduction section, and added relevance to the cited articles. This revise is marked in red in the manuscript. For example:

  • “However, the surface of SiO2 is usually rich in hydroxyl groups (Si-OH) which tend to adsorb moisture from the air and occur agglomeration, further leading to the degradation of SR composite properties [14, 15].”has simplified in the second paragraph of the introduction and clearly marked in red.
  • “Currently, surface modification has been validated as an effective method to solve agglomeration problems and improve the affinity between silica particles and matrix” has been refined and reformed in the second paragraph of the introductionand clearly marked in red.
  • “In theory, the stronger the interfacial force between particles and matrix present the better the properties of composites.” has been deleted in the second paragraph of the introduction and clearly marked in red.
  1. Fig. 1 is not suggestive. The same composites were prepared for non-functionalized silica? How the authors prepared the other composites?

Response: Thanks for your suggestion. We are sorry that we have overlooked the preparation method of SR nanocomposites for nonfunctionalized silica. The SR nanocomposites prepared for nonfunctionalized silica were fabricated in the same way as SR/f-SiO2 nanocomposites, we have added in the article and marked in red.  

  1. In the Results and discussion section there are some errors in the FTIR interpretation in the attribution of the vinyl groups. The authors need to improve this section.

Response: Thanks for your comment. In this manuscript, the error of Si=CH=CH2 bound is corrected to Si-CH=CH2 and clearly marked in red. Meanwhile, the explanation of vinyl attribution in FTIR has been confirmed by former experiments and is cited in the paper.

  1. In the particle size and BET analysis there are only a few differences between the SiO2 and f-SiO2 in therms of aggregation of the particles and their BET surface. Why the BET surface is important for the properties of the mixtures with SR and of composites? The dispersion of the particles in the in situ generated silicone matrix was not proved. Fig. 3 suggested only the hydrophilicity of the particles, the SEM images are not conclusive. SEM on the surface and in the section of the composites is needed.

Response: Thanks for your comment. In General, particles with high specific surface area have high specific surface energy, are unstable and lead to agglomeration of particles to reduce the specific surface energy. Hence, particles with higher specific surface area are poorer in terms of dispersibility. This point is added in the paper. From the data calculations in Table 1 and the TEM morphologies of f-SiO2/SR and SiO2/SR in Figure 6, it can be concluded that the modification effectively reduces the BET surface energy of the particles and exhibits better dispersion in the SR matrix. f-SiO2 dispersion in the SR matrix is demonstrated by the TEM characterization method in Figure 6.

  1. The grafting ratios must be determined from TGA data. The DSC endotherm process at 575℃ is not conclusive, at that temperature there is a mass loss in the TGA curve. So that this section needs improvements.

Response: Thanks for your comment. For the calculation of the grafting rate we have supplemented and explained the formula in the section 2.4. Characterization, and the grafting ratio in the section Results and Discussion of TG and DSC.

“The grafting ratio of the modified product is calculated by the following formula (2.1).

                  (2.1)

In the formula,  is the grafting rate of the modified product (%),  is the mass of the product after grafting (g),  is the initial mass before modification (g).” has been added in the manuscript and marked in red.

“Moreover, the grafting ratio of the f-SiO2 was calculated from formula (2.1) as 2.84%.” has been added in the paper and marked in red.

  1. Fig. 6 need some comments, the differences are not such conclusive.

Response: Thanks for your suggestion. We are sorry that this expression has confused you. Figure 6 (a, b) shows the TEM morphology of SiO2/SR composites at SiO2 10wt.% filler content, and Figure 6 (c, d) f-SiO2/SR composites at SiO2 10wt.% filler content. Here, Figure 6 (b) and (d) are the high-resolution TEM morphologies of the two composites, respectively. The detailed explanation of Figure 6 has also been revised in the article and highlighted in red.

  1. The tensile-stress curves for nonfunctionalized SiO2 based composites indicate a higher mechanical performance than the composites based on f-SiO2.

Response: Thanks for your suggestion. The rubber molecular chain has good flexibility and low interaction force. When large deformation occurs, the molecular chain network will deform, and when the external force is released, the networks will recover. This is why rubber materials has good elasticity. For f-SiO2/SR composites, the motion of SR molecular chain is greatly limited because of the strong chemical bonds between f-SiO2 particles and SR matrix. As for SiO2/SR composites, due to the relatively weak van der Waals force between SiO2 particles and SR matrix, the movement of SR molecular chain will not be greatly affected, resulting the better extensibility than that of f-SiO2/SR composites system. Although the f-SiO2/SR elongation break is lower than that of SiO2/SR, it is above that of pure SR.

We have already made detailed revisions and adjustments to the above comments. Thank you again for your valuable comments.

Reviewer 2 Report

I think that the manuscript is very well presented and it is a significant contribution
The manuscript can be accepted for publication after addressing the following minor issue
Please check the caption for figure 6 and references to it in the text. These images are SEM and not TEM because the Hitachi S-4100 microscope is a scanning microscope (https://manualzz.com/doc/57491722/hitachi-s-4100-manual)

Author Response

I think that the manuscript is very well presented and it is a significant contribution. The manuscript can be accepted for publication after addressing the following minor issue. Please check the caption for figure 6 and references to it in the text. These images are SEM and not TEM because the Hitachi S-4100 microscope is a scanning microscope (https://manualzz.com/doc/57491722/hitachi-s-4100-manual)

Response: Thank you for your comment. Your opinion is of great help to us. We are sorry for the simple error caused by an oversight. We have checked and corrected the TEM model in the manuscript. It has been corrected from "Hitachi S-4100" to "Hitachi HF5000", and marked in red.

Thank you again for your valuable comments.

Round 2

Reviewer 1 Report

The authors have improved their manuscript, therefore it can be accepted in the present form.